# Protocol for an extended scoping review on the use of virtual nominal group technique in research

Susan Humphrey-Murto[1,2]*, Seung Ho Lee[3], Michael Gottlieb[4], Tanya Horsley[5,6], Bev Shea[7], Karine Fournier[8], Christopher Tran[1], Teresa Chan[9], Timothy J. Wood[2], Olle ten Cate[10]

1 Department of Medicine, University of Ottawa, Ottawa, ON, Canada, 2 Department of Innovation in Medical Education, University of Ottawa, Ottawa, ON, Canada, 3 Faculty of Medicine, University of Ottawa, Ottawa, ON, Canada, 4 Department of Emergency Medicine, Rush University Medical Center, Chicago, IL, United States of America, 5 Royal College of Physicians and Surgeons of Canada, Research, Ottawa, ON, Canada, 6 School of Epidemiology and Public Health, University of Ottawa, Ottawa, ON, Canada, 7 School of Epidemiology and Public Health, Faculty of Medicine, University of Ottawa, Ottawa, ON, Canada, 8 Health Sciences Library, University of Ottawa, Ottawa, ON, Canada, 9 Department of Medicine, Division of Emergency Medicine; Division of Education & Innovation; Faculty of Health Sciences, McMaster University, Hamilton, ON, Canada, 10 Utrecht Center for Research and Development of Health Professions Education, Division of Education, University Medical Center Utrecht, Utrecht, Utrecht, The Netherlands

* shumphrey@toh.on.ca

## Abstract

## Introduction

Consensus group methods such as the Nominal Group Technique (NGT) and Delphi method are commonly used in research to elicit and synthesize expert opinions when evidence is lacking. Traditionally, the NGT involves a face-to-face interaction. However, due to the COVID-19 pandemic, many in-person meetings have moved to online settings. It is unclear to what extent the NGT has been undertaken in virtual settings. The overarching aim of this scoping review is to explore the use of the virtual NGT in research. Our specific objectives are to answer the following questions: To what extent has the NGT been used virtually? What modifications were made to accommodate this online format? What advantages and disadvantages were noted by authors in comparison with the face-to-face mode of the technique?

## Materials and methods

This scoping review will follow the steps outlined by Arksey and O'Malley and the PRISMA-ScR guidelines. Several pilot searches were completed to refine inclusion and exclusion criteria. Media Synchronicity Theory will provide a conceptual framework to inform the research, including data extraction and summarizing results. As an additional extension to the literature review, online interviews with corresponding authors will be conducted to gather further information.

**Funding:** This project is being funded by a Medical Education Research Grant, Department of Medicine, University of Ottawa. The funders will not have any role in the study design, data collection, and analysis, decision to publish or preparation of the manuscript.

**Competing interests:** The authors have declared that no competing interests exist.

## Introduction

Consensus group methods are commonly utilized to synthesize expert opinions when evidence is lacking or contradictory. They have been widely used in multiple disciplines including healthcare, education, engineering and management [1, 2] to inform health-related activities such as defining diagnostic criteria, classifying diseases, selecting quality indicators, informing management guidelines, and educating healthcare professionals [1–7].

The nominal group technique (NGT) was developed as a procedure to facilitate effective group decision-making in social psychological research [3]. It involves an explicitly structured format to generate and rank or prioritize ideas. The steps of the NGT are as follows: presentation of the nominal question, silent generation of ideas in writing, round-robin feedback from group members to record each idea in a succinct phrase, group discussion for clarification of ideas, individual anonymous voting or ranking on priority areas with the group decision being mathematically derived, and feedback of results followed by further discussion and re-rating [8, 9]. Hence it can be seen that the NGT adheres to the foundational principles established for consensus methods including anonymous voting, iteration, controlled feedback, statistical group response, and structured interaction [1].

One of the key features of the NGT that differentiates it from other consensus techniques (e.g. Delphi) is the inclusion of a structured face-to-face meeting, typically involving 5–12 participants [6]. The ability for participants to discuss and debate is touted as a particular strength of the method. It allows disparate ideas on matters of shared interest to not only be expressed and collated, but if differences in opinions are found the reasons can be explored. Due to its collaborative nature, the NGT may increase stakeholders' ownership of the ensuing research and likelihood of changing practice or policy [3]. Limitations include a smaller number of participants than the Delphi, and the potential for dominant participants to unduly influence the group.

The coronavirus disease of 2019 (COVID-19) pandemic has fundamentally shifted how we work, learn and conduct research [10]. The use of traditional approaches in research including the NGT have been altered; for example, some researchers have moved the face-to-face in person NGT meetings to a virtual environment [11–14]. Nelson (2022) used 3 nominal groups to identify preferred burnout strategies for residents, involving two virtual NGTs and one face-to-face meeting. A recent scoping review identified 30 methodological decision points for investigators using NGT [15] based on a review of 57 studies. Meeting face-to-face versus virtually was considered a decision point, but only one study was provided in support [16]. Another article provided recommendations for how to convert from traditional to virtual NGTs, but was solely based on the authors' experiences of running virtual NGTs [14]. Reported benefits of virtual NGTs included accessibility and lower time investment, but there were challenges with managing participants during the brainstorming phase (unable to interact directly, harder to pick up on both verbal and non-verbal cues) leading to issues with managing time [14]. Specifically, it was noted that active interaction or conversation among participants was not enhanced by the online setup.

It is unclear to what extent other researchers have transitioned the NGT to a virtual format, what types of virtual platforms are being most used and what modifications have been required when running the NGT sessions. In addition to these exploratory queries, it is important to consider whether the virtual platforms offered any challenges, or perceived advantages or disadvantages. Taken collectively, these lessons learned would be helpful to guide future virtual NGT users.

Fundamentally, the NGT is a process for structured group communication. The surge in adoption of electronic communication technologies during the pandemic has fundamentally

changed how individuals interact. A conceptual framework that may improve our understanding of how media influences communication is Media Synchronicity Theory [17]. This theory considers the effectiveness of technology to support group work. It begins by redefining tasks for all communication activities into two processes: conveyance (transmission of new information to enable the receiver to create and revise their individual understanding of a problem) and convergence (how individuals understand information and negotiate or reach a common understanding of an issue). Convergence is thought to require more rapid authentic back and forth information transmission. For the NGT, the idea generation phase would presumably require more conveyance and the discussion and clarification phase more convergence. Thus, both would be required, but the proportion may vary depending on the complexity of the research or other aspects. For example, a homogeneous group of participants with similar backgrounds and a shared mental model may require less deliberation. Other aspects to consider are visual and physical "symbols". A reduction in social presence may be noted when physical, visual, or verbal symbols are removed [17].

In summary, the NGT is a widely used technique used in research. The face-to-face interaction is fundamental to the process, but as a result of the COVID pandemic, many in-person meetings have moved to online settings. It is unclear to what extent the NGT has been undertaken in virtual settings and whether NGTs will increasingly be transitioned to virtual platforms. Since the NGT is a key method used to inform important healthcare decisions, it is imperative to develop an understanding its use in a virtual environment.

## Objectives

The overarching purpose of this study is:

To explore the use of the virtual Nominal Group Technique (virtual NGT) in research.

## Materials and methods

Since the literature on virtual NGTs is relatively new and poorly defined, a scoping review provides an appropriate method to explore and describe the breadth of knowledge related to the topic of interest. It will allow the research team to map the literature, identify key concepts, gaps in the literature and types and sources of evidence [18, 19]. Thus, a scoping review using the Arksey and O'Malley framework [20] has been initiated and will follow the PRISMA-ScR checklist for reporting Scoping Reviews [21] (See S1 File).

### Step 1: Identifying the research question

We begin with a broad question: "to explore the use of the virtual NGT in research." In an iterative process the research team further refined the context. The study population will include all English-language published research in healthcare and healthcare education which used the NGT and did so in a virtual format. The latter could include teleconferencing, videoconferencing, or any other non-face-to-face format. No comparison intervention will be required. The outcomes could include any of the following: number of items generated in the NGT, author description of use of the virtual NGT, perceived success of the process, benefits, risks, and challenges. Based on a preliminary search of the literature, the following questions will also be addressed:

Specific objectives are to answer the following questions:

To what extent has the NGT been used virtually?

What virtual communication platforms are used?

What modifications to the technique were made to accommodate this online format?

What advantages and disadvantages were noted by authors?

## Steps 2 & 3: Identifying relevant articles and article selection

We began with the following: Population, Concept, Context framework. The population for this scoping review includes any published research studies using the nominal group technique, the concept entails the use of virtual modalities to execute the nominal group technique and context may involve any study topic.

In order to identify relevant articles and further refine the inclusion and exclusion criteria several pilots were completed beginning in June 2022 in conjunction with an information specialist (KF). The first pilot established that several articles used a virtual nominal group technique yet did not mention "virtual or online" in the title or abstract, thus necessitating we remove those terms from the search strategy. (SHM) A comprehensive search completed June 28, 2022 led to the concern that all articles (n = 4116) would need to be pulled for full text review because it was not clear from title/abstract if the method was face-to-face or virtual (See S2 File). Working on the assumption that pre-pandemic authors would be more likely to mention that the NGT was not face-to-face in the title or abstract, as this would be a deviation from the norm, we completed 2 other pilots to validate this assumption. We pulled 100 references from 2015 to 2019 (pre-pandemic) and 100 from 2020–2022 (pandemic). Of the 85 relevant references screened in the "pre-pandemic pilot" (SHL), 64 full-text articles were reviewed as they did not specify the study setting, of which 5.9% (4/64) were conducted virtually. For the "pandemic pilot", 80 relevant references were screened (SHL) with 69 requiring a full-text review, which revealed that 34.8% (24/69) employed NGT virtually or with a virtual component.

Since the bulk of the virtual NGT literature were published during the COVID-19 pandemic, it was ultimately decided that the final search would be limited to the pandemic era using the official date of WHO declaration (March 11, 2020) [22]. The team agreed that for any relevant abstracts that do not clearly describe a study setting, full-text articles would be reviewed.

The final search strategies were developed by an information specialist (KF) and peer reviewed using the PRESS guideline [23]. The searches were conducted July 15th, 2022 in: MEDLINE(R) ALL (OvidSP), Embase (OvidSP), CINAHL (EBSCOHost), ERIC (OvidSP), Education Source (EBSCOHost), APA PsycInfo (OvidSP), Web of Science, and Scopus to retrieve references published in 2020 to July 2022. Drafting the search strategy was also informed by a scoping review for the concept of NGT [15]. No search filters or language limit were used, but conference abstracts were removed when feasible since only full papers are of interest. The search strategies are included in S3 File. Final Search Strategy and the final output was exported to Covidence. The electronic search of the databases identified 2,589 citations 1,656 duplicate records were removed using Covidence (Veritas Health Information, Melbourne, Australia), which left 933 references for the screening phase.

Abstract and title review will be completed in duplicate and any conflicts resolved by a third team member. Inclusion and exclusion criteria are listed in Table 1. The same process will be followed for screening of full text articles. Final inclusion and exclusion criteria are listed in Table 2.

**Table 1.  Inclusion and exclusion criteria at title and abstract stage.**

|  | Inclusion Criteria | Exclusion Criteria |
|---|---|---|
| Articles to be pulled for full review | Any article that mentions nominal group technique/method/ consensus nominal AND any use of alternative face-to-face (e.g., conference video, telephone, Zoom, online) OR any article where it is unclear if virtual or face-to-face | No consensus method noted |
| | | Delphi as the only consensus method noted |
| | Any research topic | Focus group as the only method noted |
| | English language | |

**Table 2. Final inclusion and exclusion criteria.**

| Inclusion Criteria | Exclusion Criteria |
|---|---|
| English Language | Nominal group technique lacking any of the 4 stages |
| Date limit: January 1, 2020 –July 15, 2022 | Conference proceedings, published abstracts, reviews, editorials, opinion pieces |
| Full text articles | |
| Original research using the nominal group technique | |
| Nominal group technique must be described in sufficient detail (e.g. cannot simply mention "nominal group technique" with no further description) | |
| Must mention that all 4 key stages of the nominal group: idea generation, sharing of ideas, discussion/clarification and voting were completed | |
| Could be one of several consensus methods used within a single study | |
| May include any research topic | |
| All stages of the nominal group technique were completed virtually (any of e-mail, online, any virtual platform, telephone) | |
| Can combine virtual and face-to-face for any given stage | |

## Step 4: Charting the data

Based on our review of the literature, a preliminary data extraction form will be created, and this will be refined after 3 or more team members have piloted on several articles. Demographic information will be collected such as publication year, journal, research topic, whether seeking consensus at the local, national or international level, country, date the NGT was carried out, and purpose of the study. We will seek information on the NGT itself such as process for item generation, number of participants and moderators, geographic distribution and heterogeneity of participants, number of items generated and included in the final list, number of rounds, formal feedback to participants, and the definition of consensus. We will extract information on the virtual platforms used, and consider concepts related to the Media Synchronicity Theory such as transmission velocity (e.g., did the authors comment on technical issues?), parallelism (e.g., was there a simultaneous transmission of information via chat functions?), and symbol sets (e.g., was there visual as well as auditory transmission?). We will explore what modifications were made to each stage of the NGT to accommodate the virtual environment and any descriptions of benefits or challenges described by the authors.

Through regular meetings, the data extraction form will be refined through an iterative process where items can be added or deleted, and definitions for each type of data to be extracted will be clarified. The revised extraction form will be piloted on 6 articles reviewed by 2–4 members of the research team. Further meetings will be held until there is consensus on a final form. At least two members will fully and independently review each article by applying the final extraction form on 20% of articles. Thereafter, one member of the research team will carry out data extraction with verification from a second member. Any ambiguous items that arise will be resolved by the PI and senior investigator.

## Step 5: Collating, summarizing, and reporting the results

Both quantitative and thematic analyses will be used to synthesize study results. Quantitative analysis will focus on the nature (e.g. education, clinical research, guideline development) and distribution of relevant articles. Three members of the research team will independently review the data to identify preliminary themes as informed by the Media Synchronicity Theory.

Group meetings with all team members will be held to review all available data and to agree on a final summary of findings.

### Step 6: Survey corresponding authors

Based on our preliminary review of a few studies, most authors did not directly address benefits and challenges of moving from face-to-face to virtual settings to conduct the nominal group session. This may reflect journal word count limits. We therefore plan to send an online survey to corresponding authors of included articles. The survey would seek to confirm which virtual platform was used and for which steps of the NGT, if additional functions were used such as chats, modifications made to the NGT to accommodate the virtual platform, why the virtual platform was used, their general impressions and perceived benefits and challenges, comparing their experience with face-to-face NGTs and any lessons learned. The survey can be found in S4 File. Online survey of corresponding authors for included studies. Written ethics approval has been granted from the University of Ottawa Research Ethics Board on September 27, 2022. Written informed consent was obtained from all participants who were included in the survey. Since this is an additional, and non-mandatory step in the Scoping review, we would consider that a response of at least one half of participants would add valuable information to the paper. We plan to disseminate our findings in a peer reviewed publication, as well as present the results at local, national and international conferences. The protocol is not currently registered.

## Discussion

The strengths and limitations of this study are as follows: this will be the first scoping review exploring the use of virtual platforms to perform the Nominal Group Technique (NGT). One key feature of the traditional NGT is the face-to-face meeting, but since the COVID-19 pandemic, many researchers have pivoted to use online modalities. The identification and data synthesis will involve several databases; MEDLINE(R) ALL (OvidSP), Embase (OvidSP), CINAHL (EBSCOHost), ERIC (OvidSP), Education Source (EBSCOHost), APA PsycInfo (OvidSP), Web of Science, and Scopus to reduce the risk of potential missed publications. This study will adhere to the steps outlined by Arksey and O'Malley and the PRISMA-SCR checklist with additional surveys of corresponding authors to provide richer data. This study will be limited to English-language studies which may limit generalizability. This study will be restricted to the English language and cover a restricted timeframe which may bias results. Published research articles may not elaborate on benefits and challenges of pivoting to the use of the virtual NGT, so this may limit the richness of the data collected. Our team speculates that authors will report that some aspects of the virtual NGT will be positive, such as the ease to convene a group of geographical spread experts, the ability to do different NGTs on the same day and the ease of recording. Studying the NGT in a virtual format will not only increase our understanding of the method but has the potential to inform rigorous use and best practices, something that has been noted to be lacking in the consensus method literature [6, 24, 25]. Ethics has been acquired. Since the NGT is widely used to inform decisions in multiple disciplines and online meetings have become common practice, it is imperative to develop a better understanding of the use of the NGT in a virtual environment. Results will be disseminated through peer-reviewed publication and presentations at national and international meetings.

## Supporting information

**S1 File. PRISMA-SCR checklist.**
(DOCX)

**S2 File. Search methods.**
(DOCX)

**S3 File. Final search strategy.**
(DOCX)

**S4 File. Online survey of corresponding authors for included studies.**
(DOCX)

**S5 File. PRISMA-P checklist.**
(DOC)

## Acknowledgments

Dr. Humphrey-Murto would like to acknowledge the work of Amanda Pace and Kate Scowcroft from the Department of Innovation in Medical Education for assisting with the study. In addition, Dr. Humphrey-Murto would like to acknowledge the Faculty of Medicine and Department of Medicine, University of Ottawa for the Tier 2 Research Chair.

## Author Contributions

**Conceptualization:** Susan Humphrey-Murto, Seung Ho Lee, Michael Gottlieb, Tanya Horsley, Bev Shea, Karine Fournier, Christopher Tran, Teresa Chan, Timothy J. Wood, Olle ten Cate.

**Data curation:** Susan Humphrey-Murto, Seung Ho Lee, Michael Gottlieb, Tanya Horsley, Bev Shea, Karine Fournier, Christopher Tran, Teresa Chan, Olle ten Cate.

**Formal analysis:** Susan Humphrey-Murto, Seung Ho Lee, Michael Gottlieb, Tanya Horsley, Bev Shea, Karine Fournier, Christopher Tran, Teresa Chan, Timothy J. Wood, Olle ten Cate.

**Funding acquisition:** Susan Humphrey-Murto, Seung Ho Lee, Michael Gottlieb, Tanya Horsley, Bev Shea, Karine Fournier, Christopher Tran, Teresa Chan, Timothy J. Wood, Olle ten Cate.

**Methodology:** Susan Humphrey-Murto, Seung Ho Lee, Michael Gottlieb, Tanya Horsley, Bev Shea, Karine Fournier, Christopher Tran, Teresa Chan, Timothy J. Wood, Olle ten Cate.

**Project administration:** Susan Humphrey-Murto.

**Validation:** Susan Humphrey-Murto, Seung Ho Lee, Michael Gottlieb, Tanya Horsley, Bev Shea, Karine Fournier, Christopher Tran, Teresa Chan, Timothy J. Wood, Olle ten Cate.

**Writing – original draft:** Susan Humphrey-Murto.

**Writing – review & editing:** Susan Humphrey-Murto, Seung Ho Lee, Michael Gottlieb, Tanya Horsley, Bev Shea, Karine Fournier, Christopher Tran, Teresa Chan, Timothy J. Wood, Olle ten Cate.

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
