## [Decision Letter · Decision Letter 0]

18 Nov 2022

PONE-D-22-27748Protocol for an Extended Scoping Review on the use of Virtual Nominal Group Technique in ResearchPLOS ONE

Dear Dr. Humphrey-Murto,

Thank you for submitting your manuscript to PLOS ONE. After careful consideration, we feel that it has merit but does not fully meet PLOS ONE’s publication criteria as it currently stands. Therefore, we invite you to submit a revised version of the manuscript that addresses the points raised during the review process.

We look forward to receiving your revised manuscript.

Kind regards,

Ernesto Iadanza

Academic Editor

PLOS ONE

Journal Requirements:

5. We note that this manuscript is a systematic review or meta-analysis; our author guidelines therefore require that you use PRISMA guidance to help improve reporting quality of this type of study. Please upload copies of the completed PRISMA checklist as Supporting Information with a file name “PRISMA checklist”.

Reviewers' comments:

Reviewer's Responses to Questions

**Comments to the Author**

1. Does the manuscript provide a valid rationale for the proposed study, with clearly identified and justified research questions?

Reviewer #1: Yes

Reviewer #2: Yes

2. Is the protocol technically sound and planned in a manner that will lead to a meaningful outcome and allow testing the stated hypotheses?

Reviewer #1: Yes

Reviewer #2: Yes

3. Is the methodology feasible and described in sufficient detail to allow the work to be replicable?

Reviewer #1: Yes

Reviewer #2: Yes

4. Have the authors described where all data underlying the findings will be made available when the study is complete?

Reviewer #1: Yes

Reviewer #2: Yes

5. Is the manuscript presented in an intelligible fashion and written in standard English?

Reviewer #1: Yes

Reviewer #2: Yes

6. Review Comments to the Author

You may also provide optional suggestions and comments to authors that they might find helpful in planning their study.

Reviewer #1: The protocol addresses a relevant topic.

Each step/decision has been clearly motivated as the in case of the data limit.

I suggest to specify (if any) a minumum number of responders has been define for the survey,

In the same way, specify if a subgroup analysis is expected for some specific research topics, being the inclusion criteria is quite wide (any research topic).

Finally, Appendix 5 is not mentioned in the paper neither included in the list of supporting information (pag. 13).

Reviewer #2: The article is very interesting and proposes a topic inherent the research methodology that is really timing. In addition, a study of the nominal group technique (NGT) is something extremely original and worthy of publication. The structure of the paper is consistent and correct and the English is fluent and clear.

7. PLOS authors have the option to publish the peer review history of their article (what does this mean?). If published, this will include your full peer review and any attached files.

Reviewer #1: **Yes: **Di Bidino Rossella

Reviewer #2: No

---

## [Author Response · Author response to Decision Letter 0]

5 Jan 2023

Response to Reviewers letter has been attached (also written below). A revisions table has also been uploaded in the "attach files" section. Please see both for further details! 

Dear Dr. Ernesto Iadanza, 

RE: PONE-D-22-27748 

Protocol for an Extended Scoping Review on the use of Virtual Nominal Group Technique in Research

Thank you for providing us an opportunity to re-submit our manuscript. As part of the re-submission, please note our error in the authorship order, it has been revised. The response to reviewers has been uploaded as a separate word document titled “Revisions table response to reviewers”. 

Regarding the other items:

1. Style has been verified to match the PLOS one requirements. Please note that the Supplemental Files have been re-uploaded with the correct PLOS one title requirements as well. 

2. We have updated the grant information (our local grants do not have “grant numbers” so not included). The financial disclosure now matches the funding information.

3. Upon reflection we have changed our response to the repository information for data to meet requirements of our ethics board. “The entirety of the data collected and analyzed for this review will be within the manuscript and its Supporting Information files. Interview data will be made available upon request and approval through the University of Ottawa Research Ethics board.” 

4. Full ethics statement provided in the manuscript under methods.

5.Uploaded copies of the marked-up manuscript with track changes titled “Revised Manuscript with Track Changes” and an unmarked version of the manuscript titled “Manuscript”. Please note that this manuscript is a scoping review and two PRISMA Checklists have been uploaded as Supplemental Files (S1_PRISMA SCR Checklist and S5_PRISMA-P Checklist)

6. The reference list has been analyzed and follows PLOS one’s referencing guidelines. 

We look forward to your response.

Sincerely,

Susan Humphrey-Murto 

on behalf of the research team

---

## [Editor Report · Decision Letter 1]

8 Jan 2023

Protocol for an Extended Scoping Review on the use of Virtual Nominal Group Technique in Research

PONE-D-22-27748R1

Dear Dr. Humphrey-Murto,

We’re pleased to inform you that your manuscript has been judged scientifically suitable for publication and will be formally accepted for publication once it meets all outstanding technical requirements.

Kind regards,

Ernesto Iadanza

Academic Editor

PLOS ONE

Additional Editor Comments (optional):

Reviewers' comments:

<quillbot-extension-portal></quillbot-extension-portal>

---

## [Editor Report · Acceptance letter]

11 Jan 2023

PONE-D-22-27748R1 

Protocol for an extended scoping review on the use of virtual nominal group technique in research 

Dear Dr. Humphrey-Murto:

I'm pleased to inform you that your manuscript has been deemed suitable for publication in PLOS ONE. Congratulations! Your manuscript is now with our production department. 

Kind regards, 

on behalf of

Dr. Ernesto Iadanza 

Academic Editor

PLOS ONE